# A qualitative study on the challenges of Afghan child labourers in Tehran

**Arash Ziapour**[1], **Fakhreddin Chaboksavar**[2], **Ahmad Ahmadi**[3], **Javad Yoosefi Lebni**[4]*, **Neda Kianipour**[5], **Parisa Janjani**[1], **Nafiul Mehedi**[6]

**1** Cardiovascular Research Center, Health Institute, Imam-Ali Hospital, Kermanshah University of Medical Sciences, Kermanshah, Iran, **2** Nursing Care Research Center, Health Research Institute, Babol University of Medical Sciences, Babol, I.R. Iran, **3** Allameh Tabataba'i University, Tehran, Iran, **4** Social Determinants of Health Research Center, Lorestan University of Medical Sciences, Khorramabad, Iran, **5** Students Research Committee, Kermanshah University of Medical Sciences, Kermanshah, Iran, **6** Shahjalal University of Science and Technology, Sylhet, Bangladesh

* j.yousefi28@yahoo.com

**Data Availability Statement:** "The data are not publicly available due to their containing information that could compromise the privacy of research participants as the data contain potentially identifying child labourers information. Interview

## Abstract

Afghan refugees child labourers face many challenges as they are labelled as refugees. In an attempt to explore these challenges, the present study was conducted in Tehran with a qualitative approach. The present qualitative research was conducted using a conventional content analysis approach with 25 Afghan child labourers in 2022 in Tehran. Child labourers were selected through purposive and snowball sampling and interviewed in a semi-in-depth interview. The data were analysed based on Granheim and Lundman's criteria, as well as those of Guba and Lincoln, to further enrich the findings. A total of 3 categories, 13 sub-categories, and 183 initial codes were extracted from the data analysis, including: "psychological challenges" (history of harassment and abuse, negative effects, high-risk behaviours, and family detachment); "health challenges" (physical problems, inappropriate accommodation, medical/therapeutic problems, and health threats); and "social challenges" (neglected childhood, dual identity, educational limitations, inadequate social support, social isolation, and social humiliation). At the individual level, it is possible to meet the child labourers' health needs and make them aware of the hazards of working in the streets through physical examination at certain intervals and holding training workshops on harassment prevention, anger control, prevention of high-risk behaviours, prevention of infectious diseases such as hepatitis, and strengthening self-confidence to improve health. Also, the Afghan child labourers' work could be intervened at the social and familial level by providing accommodation in more suitable neighbourhoods, providing health insurance for child labourers, creating the chances of studying in schools, preventing dropouts, and strengthening social relations in order to improve children's health.

## Introduction

With the growing number of refugees around the world, evaluating the refugees' health becomes more important in determining the health state of society [1]. An increase in the

guide and Complete list of study themes and representative quotes is included as supplementary material. It is required that all researchers who wanted to work on this data submit their data request access to research_it@kums.ac.ir (Deputy of Research and Technology)."

**Funding:** The authors received no specific funding for this work.

**Competing interests:** The authors declare that they have no competing interests.

number of migrant refugees is a global issue that has affected many countries [2]. According to the facts and figures reported by the International Labour Organisation (ILO), around the world, about 218 million children between the ages of 5 and 17 are forced to do some kind of manual work, whether paid or unpaid [3]. In Iran, the number of child labourers is 1,200,000, of whom 35,000 reside in Tehran. This population is increasing with global population growth, migration, and the increasing rate of urbanisation [4–6].

Most of the refugees in Iran are from Afghanistan [2]. Political and economic issues have been raised as the most common causes of Afghan migration to Iran within the past 37 years [7]. The results of a census in 2015 [8] indicated the presence of 1,759,448 formally registered refugees, including 1,583,979 Afghans in Iran. The same source also reported that Tehran was a city with the largest population of formal refugees. At the same time, according to informal statistics, the total number of Afghans in Iran, including unofficially registered refugees, is estimated to be close to three million [9].

The relatively low level of human capital of refugees and the legal restrictions on many Afghans in Tehran have placed them among the lowest classes of society [7]. Sanitation facilities, in turn, can cause many health deficiencies, especially among children. Such families mainly live in slums, and their children tend to work in areas such as subways, warehouses, grocery stores, tailors, supermarkets, farms, and even at home to make goods manually [10]. The increasing rate of child laborer is so high that it has affected both developed and developing societies [11]. The issue of child labor is mainly caused by the parents' unemployment and poverty in a wide range [12].

According to the latest definitions, children and young people who live and work in the streets are known as child labourers [13]. Work deprives children of their basic rights. Their childhood is gone, and their basic needs such as education, health, and recreation are not met [14–16]. According to the recent definition of Human Rights Watch, street child labourers are under the age of 18. They work and live on the streets in different ways. In fact, these children spend most of the day and night in the street. They are mostly forced to work in the streets or unsafe places to earn a living [17, 18] They are either the breadwinner or work to meet their personal needs [19]. There is sample evidence that the most common reasons for child labourers to live on the streets are poverty and the abusive behaviour of alcoholic parents [20]. A similar report also showed that the parents' deaths and unhealthy family relationships are among the causes that force child labourers to stay on the street [21]. Meanwhile, working and living outside the home is risky for them. For example, many studies in different countries have shown that the current conditions of child labourers, their jobs, and what makes them end up in the streets include adverse physical, mental, and sexual health consequences and social risks [11, 22–25]. According to a body of research, among the major health issues affecting child labourers are physical injuries, HIV/AIDS, sexually transmitted diseases, sexual and reproductive health disorders, harassment and sexual harassment, drug abuse, and mental problems. Due to extreme poverty and limited access to healthcare and education, child labourers are considered the most vulnerable population in society. They are the victims of violence and are forced to live in the streets, scavenging, begging, and peddling in the nearby slums in polluted environments [19, 26, 27].

Child labour has been linked with significant and long-term health impacts. The results of a study by Banstola et al. showed that half of the children working in brick factories in Nepal had difficulty breathing, and more than half of them suffered from musculoskeletal disorders of grade III BPD with severe pain in the elbows, hands, wrists, knees, ankles, and feet. 60% of the child labourers suffered from insomnia. About 6.7% were involved in smoking, and 21.8% consumed alcohol [28]. The findings reported by Betul and Bilal showed that child labour causes stress, frustration, aggression, and depression among children [29]. The health state of

child labourers in another study by Abate et al. in Eastern Ethiopia showed that most participants (92.73%) regularly consumed alcohol. Depression (39.22%) and peer pressure (43.14%) were the most common causes of alcohol consumption [19]. Other similar scientific evidence has shown that in Ethiopia, more than four million children live in very difficult conditions [30, 31]. They are at high risk of sexual and physical abuse [32]. Other evidence indicates that 15.6% of child labourers engage in risky sexual activity and 61.6% face health issues [19, 33].

In recent decades, authorities and researchers in Iran have emphasised the issue of child labourers in the streets [17]. Dealing with the problem of child labourers is of particular importance. First of all, the working issue and child labour, regardless of the facts and figures, are important social matters worthy of attention in society and have many social effects. Second, child labourers are among the most deprived children who do not benefit from the most basic rights. Third, because of their special living conditions, child labourers are faced with many wrong and risky behaviours that not only seriously harm the child but also threaten the health of society and other people. So far, few studies have been conducted on child labourers, and they mostly investigated the underlying causes quantitatively. Less attention has been paid to their health problems from different angles. Also, the challenges of Afghan child labourers, who can be different from other child labourers because they are refugees, have not been investigated so far. Thus, it is essential to conduct qualitative research to access the deep layers of their lives and clarify the basic problems they face. Therefore, the present study aimed to explore the challenges facing Afghan child labourers in Tehran via a qualitative approach.

## Methods

### Study design

The present research was conducted with a qualitative approach using conventional content analysis among Afghan child labourers in Tehran in 2022. The knowledge produced by the content analysis method is based on the participants' unique experiences and the actual data. In qualitative content analysis, the goal is to categorise the information obtained from interviews and observation protocols. In this method, the information is collected directly from the participants, and the predetermined categories and theoretical views of the researcher are not analysed. Codes and categories are extracted from raw data inductively [34]. In a content analysis, the researcher explores the recurrence of an event, the meaning and connections of the words and concepts of the text, and then extract the themes from the text, the themes meant by the author, the audience, and even the culture and time to which the words and concepts belong [35].

### Participants

The research population was all Afghan child labourers working in different parts of Tehran. The inclusion criteria were having Afghan parents, being 15 years old or younger, having working experience in Iran, and being willing to participate in the study. The exclusion criteria included unwillingness to record the interview process and failure to complete the interview.

### Data collection

The participants were initially selected in a purposive sampling (n = 18) and then in a snowball sampling (n = 7). Thus, different areas of Tehran were visited to find Afghan children working in the streets and intersections. They were asked to participate in the study. After each interview, the participants were asked to introduce to the researcher other people they knew and those who met the inclusion criteria to participate in the study.

**Table 1. Semi-structured interview questions.**

| # | Item |
|---|------|
| 1 | How do you feel about working in Iran as a child manual worker? |
| 2 | What sorts of problems have you experienced so far while working here? |
| 3 | How do people react to you working as an Afghan child here? |
| 4 | How do your employer and co-workers treat you in your workplace? |
| 5 | What do you think about the government and the social security organisations in Iran? |
| 6 | What are the main challenges for Afghan child labourers in Iran? |

The data collection method in the present study was a semi-structured interview conducted face-to-face with the participants. The interview questions were formulated after three meetings among the research team members with the consensus of all researchers, each from a different speciality. The research questions were designed by the authors of the article, who have expertise in various fields including sociology, psychology, health education, and others.

The questions were finalised after a pilot phase with four participants (Table 1). Initially, based on a review of previous research studies, the authors of the article collaboratively designed four interview questions for the participants. These questions were then experimentally implemented with four participants who met the inclusion criteria for the study. Subsequently, the interviews were analysed, one question was removed, and three additional questions were added to the interviews. Finally, the interview guide was finalised with six final questions.

For the male participants, interviews were conducted by a man with a doctorate degree in health education and promotion and familiarity with semi-structured interviews. For female participants, interviews were conducted by a woman with a master's degree in health education and familiarity with research procedures. In addition to the interview guide questions, other questions were also asked to understand more about the participants' experiences and obtain more in-depth data. At the beginning of each interview, to establish proper communication and more trust, the researcher provided information about his field of study, job, and role in the research. Then he explained the objectives of the study and the procedure, as well as how to report the findings, and if the participants provided written consent, the interview began. First, some demographic questions were asked, followed by the interview guide questions and other minor questions. Three of the parents did not allow their children to participate in the research at first, but by referring to them again and explaining the research process, their consent was obtained.

Notes were also taken during the interview so that the tone of voice, pronunciation of words, laughter, crying, and pauses of the speech, as well as the physical effects of harassment, could also be recorded during the interview. All the interviews were recorded using the audio recorder device. The time and place of the interviews were determined by the participants. As the participants preferred, most interviews were held in a public place, such as a park or a cultural centre affiliated with the town hall. During the interview, no one else was present except for the researcher and the interviewee. Each interview continued until data saturation was finally achieved with 25 participants.

## Data analysis

The process of data categorization and adjustment was done in MAXQDA-2020, and its analysis was done according to the Granheim and Lundman method [36]. In the first step, after the first interview, the researcher typed the interviews in Word 2010 immediately on

the same day with the help of another research colleague. As the second step, the interview content was carefully read repeatedly by the researchers to get an overall understanding of the content. In the third step, all the interview transcripts were read line by line, word by word, with great care and patience, and the keywords that could be helpful in data analysis were selected as codes. The initial codes were extracted accordingly. In the fourth step, codes that were similar in meaning and concept could be placed in the same sub-category. Then the codes and categories were placed in the main categories, which were conceptually more comprehensive and abstract, and the themes were extracted. Finally, in a joint meeting, the entire process of data analysis was shared, and the opinions of all members of the research team were included.

## Trustworthiness

To improve the quality of the findings, Guba and Lincoln's criteria were used [37]. To increase the credibility of the research, at the end of each interview, the researcher's general understanding of the participants' words was briefly explained and approved. Also, at the end of the data coding and analysis process, the table of categories, subcategories, and codes along with the excerpts was provided to several participants (n = 7) who were literate to determine whether the researchers reported their experiences correctly or not and finally got their approval. To measure confirmability, the data analysis and findings were sent to two prominent researchers in qualitative research as well as three experts who were experienced in research on Afghan child labourers. The necessary corrections were made, and finally, the content was confirmed. To increase dependability, all the contributors to the paper were included in the process of data coding and analysis. They cast their opinions in the meetings that were held, and finally, the titles of categories and subcategories were finalised upon the approval of all the authors. To increase the transferability, a complete description of the entire research process was provided, the excerpts of the participants' interviews were provided directly and in a large volume, and the research findings were made available to several Afghan child labourers who met the inclusion criteria for this study yet did not participate. They were asked if the experiences of the participants in the study were similar to their actual experiences and whether the experiences of the participants reflected the challenges of their actual lives in Iran or not. In the end, the findings were confirmed.

## Ethical considerations

This study is the result of research project No. 4010611 approved by the Student Research Committee of Kermanshah University of Medical Sciences, Kermanshah, Iran. The authors would like to thank all the participants who patiently participated. Also, This present project received a code of ethics from Kermanshah University of Medical Sciences (IR.KUMS. REC.1401.347), clarifying the objectives of the study and the data collection procedure for the participants. The participants were assured of the confidentiality of the information they provided, such as name and address, place of residence, etc., written informed consent to participate in the study, having the interview audio-recorded, and having the right to withdraw from the study whenever they wanted. Also, due to the participants' young age, written consent was obtained from one of their parents. The process for obtaining informed consent was as follows: Upon approaching Afghan child labourers, the research process was thoroughly explained to the participants, and the consent form was read to them and fully explained. It was emphasised that participation in the study was voluntary, and they had the option to discontinue the interview at any time or choose a different time and place. Additionally, they were asked to contact one of their parents so that the researchers could meet with them and gain their consent for

their children's participation in the study, providing a detailed explanation of the research process and how the results would be disseminated. During the interview, only the researcher and the children were present, and the parents were not present at the interview location. For some participants, a meal was prepared and provided. Additionally, for some others, the researcher purchased some of the items they were selling.

## Results

A total number of 25 Afghan child laborers participated in the present study (Table 2). Also, 3 categories, 13 subcategories and 183 primary codes were extracted from the data analysis (Table 3).

### 1. Psychological challenges

The first category derived from the data was psychological challenges, which showed that Afghan child labourers have many psychological issues, such as experiencing violence and abuse, having negative emotions, and having risky behaviours. They face separation from their families, which can threaten their mental health.

**History of harassment or abuse.** A constant challenge for Afghan child labourers was the experience of harassment and abuse. Most Afghan child labourers were harassed verbally every day, especially those selling things at crossroads, who were often insulted by drivers. In some cases, though the children cleaned the drivers' cars, they received no money in return. In some other cases, girls were sexually abused, especially when there was scarcely anyone around in the streets. For the same reason, girls worked shorter hours and went home earlier to be less abused. They were asked not to be around when the street is not crowded and there is a possibility of abuse. Also, Afghan children were abused by municipal officers due to their illegal presence in Iran, and in some cases, their belongings were broken by the officers, and they could not sue because they had no legal rights here in Iran. Also, most child labourers had the experience of being bullied and robbed of all their money. This was especially the case with child labourers who worked in the poor neighbourhoods of the city. Here are a number of excerpts:

**Table 2. The child laborer participants' demographic information.**

| Variable | Level | frequency |
|---|---|---|
| Sex | Female | 10 |
| | Male | 15 |
| Age | < 10 years | 8 |
| | 10–15 | 17 |
| Education | Undergraduate | 6 |
| | Drop-out | 7 |
| | Uneducated | 12 |
| History of sex harassment | Yes | 17 |
| | No | 8 |
| Employment type | Street vendors | 18 |
| | Scavenging | 4 |
| | Other(s) | 3 |
| Birth place | Iran | 18 |
| | Afghanistan | 7 |

**Table 3. Codes, sub-categories and categories extracted from interviews with Afghan child laborers.**

| Categories | Sub-categories | Codes |
|---|---|---|
| Psychological challenges | History of harassment or abuse | Experiencing verbal harassment, mental harassment, physical and sexual abuse in the street, being abused as an Afghan, being beaten by municipal officials, blackmailing informal foremen, being robbed of income |
| | Negative effects | Jealousy towards peers, revenge and hostility towards the rich, instability and restlessness, mistrust of others, anxiety, a sense of insecurity and fear, fear of deport to Afghanistan |
| | High-risk behaviours | aggressiveness, theft, suicide, drug abuse or transport, and high-risk sexual behaviours |
| | Family detachment | Less time at home, no strong emotional bond with parents, economic independence from the family, lack of parental supervision, perceived redundancy of parents, tension and conflict with parents |
| Health-related challenges | Physical problems | Physical weakness, sunburn, cold, back pain, sore toes, pain in hands and feet, digestive problems, skin problems |
| | Inappropriate accommodation | Living in poor and high-risk areas (buying and selling drugs nearby), inadequate number of rooms, housing large populations in small and unsanitary areas, several families living in one small area, failure to maintain hygiene at home |
| | Medical and therapeutic problems | No health insurance, high cost of medication in Iran, the health staff's inappropriate behaviours towards Afghans, no vaccination |
| | Health threat | Exposure to infection (AIDS, hepatitis, COVID-19, etc.), improper nutrition, exposure to adverse weather conditions, car accidents, exposure to accidents |
| Social challenges | Neglected childhood | No fun, toy, or amusement; no spare time to spend with friends; no exercise; no time to play; early marriage (for girls) |
| | Dual identity | No enthusiasm for and interest in the homeland (Afghanistan), failure to be accepted in Iran, lack of identity certificates, sense of alienation |
| | Educational problems | Restrictions on enrollment in schools due to illegal attendance, few schools for Afghan children, refusal of some schools to enrol Afghan children, negative attitude of school administrators and students towards Afghan children |
| | Inadequate social support | Lack of supportive social organisations, lack of coherent programmes to improve children's health, lack of appropriate educational plans for child labourers |
| | Social isolation | Lack of social relations, isolation (no contact with Iranian families), loneliness, and not enough friends due to the change of work position and inability to communicate with Iranian children |
| | Social humiliation | Humiliation in the street, humiliation at work, inappropriate nicknames because of being an refugees, being socially labelled |

"*There is no single day without a fight with them. They keep swearing at me or even beating me.*"

(*10-year-old boy*)

"*Many times, as soon as the drivers find out we are Afghans, they insult us and begin to swear.*"

(*11-year-old boy*)

"*Some drivers hit us on purpose or make us work and tell us to clean the window, but as soon as we clean, they fail to pay and just drive away.*"

(*13-year-old boy*)

"*It happened several times that the municipal officials came and took our belongings with them or broke everything on the spot, but there was nothing we could do about it.*"

(*9-year-old girl*)

*Once, when the street was quiet, someone in a car called me and said, Come here; I'd like to buy chewing gum from you. As soon as I approached, he grabbed my hand and forced his hand under my clothes and*

(*12-year-old girl*)

"*So far, it has happened many times that I worked till night and collected money, but bullies came at night and took my money away and even beat me.*"

(*14-year-old boy*)

**Negative effects.** Another challenge for Afghan child labourers was developing negative emotions. When most of these children saw children of their own age in good conditions, they could feel jealous, so sometimes this sense of jealousy could gradually lead to aggression. Also, most of the child labourers had a sense of pessimism and mistrust towards others due to their many experiences of harassment. Some child labourers, especially those who were in Iran illegally, always lived with a sense of fear and insecurity because they were afraid they would be arrested by the agents and returned to Afghanistan. Depression and other mental illnesses emerged in these children. Compare the following comments by some participants:

"*When I see other children of my age who are in billion-dollar cars and everything is provided for them, I feel sad and jealous of them.*"

(*8-year-old boy*)

"*Sometimes I like to scratch and damage the expensive cars I wash.*"

(*7-year-old boy*)

"*I am a girl. I am always afraid that someone hurts me or arrests me and sends me back to Afghanistan.*"

(*9-year-old girl*)

"*I got so annoyed that I don't trust anyone anymore; even sometimes, when someone wants to be kind to me and give me some food or something, I'm afraid to approach him because I tell myself he may hurt me.*"

(*11-year-old girl*)

**High-risk behaviours.** Some child labourers were forced to deviate by the current conditions and showed aggressive behaviours. They fought with many people every day. Some said they had experienced minor thefts of phones and wallets. Still, some were abused by criminals and used to transport and sell drugs because the police were less suspicious of them due to their young age and they could do it more easily. The same issue can lead them further towards deviance, so they may join criminal gangs and become professional criminals. Some children who lived in groups in slums and collective houses under the supervision of a superior force were prone to health-threatening sexual intercourse, and such sex affairs could significantly threaten their lives and be followed by infectious diseases.

"*I have a row at least once or twice a day. Is it possible to work at a crossroads and not fight? Sometimes I fight with a driver, sometimes with other people.*"

(*14-year-old boy*)

"*Sometimes I lift a phone when I get a chance.*"

(*8-year-old girl*)

"*I often get tired of life. I was going to commit suicide once or twice. I cut a vein in my hand once, but I didn't die.*"

(*13-year-old boy*)

"*It happened several times when I was offered some money and asked to deliver a package of drugs. I did so, though I knew the hazards. I only agreed just because they paid a lot of money.*"

(*8-year-old girl*)

"*I have been smoking cigarettes and hash for years.*"

(*14-year-old boy*)

"*Several times, I had sex with a neighbour's girl, who is also Afghan.*"

(*13-year-old boy*)

**Family detachment.** Another challenge for child labourers was that they spent most of their time away from home, and this caused them not to be able to establish a strong emotional relationship with their parents, especially their father. Those who were economically independent were no longer financially dependent on the family. This lack of dependence caused the family not to be able to monitor the child's behaviour adequately. In some cases, the child had a row with the parents about how to spend the earned income.

"*I don't have strong feelings for my parents because I never see them properly. Every morning, I leave home and am in the streets until night, when I go back home. My father is a doorman, and we do not see him at all.*"

(*10-year-old girl*)

"*When I think that my parents gave birth to me in these terrible conditions, I don't feel good about them. I feel they were selfish for giving birth to me, so I can't really love them.*"

(*13-year-old girl*)

"*I don't need my parents; I make money myself. Often I don't tell them where I work and what I do.*"

(*11-year-old boy*)

"*Sometimes I don't go home for two weeks, and I stay with a friend, and I don't tell anyone where I am.*"

(*14-year-old boy*)

"*My father wants to give him all the money I earn, and sometimes I hide it from him. We have argued several times over this issue.*"

(*9-year-old boy*)

## 2. Health challenges

Afghan child labourers face many health issues such as physical problems, inappropriate housing, medical problems, health threats, and neglected childhood, which can endanger their health or even be fatal.

**Physical problems.**   Afghan child labourers are prone to sunburn in the summer and frostbite in the winter because they spend long hours of the day outdoors without any protective clothes. Also, child labourers who work on the road all day long and have no rest suffer from physical weakness, and because they do not have proper nutrition, this physical weakness becomes more severe, and they may even suffer from digestive problems. In some cases, because they stay out for long hours of the day and night and do not have proper shoes, their feet get sore. Scavenging by some other children causes injuries in them that can threaten their health.

"*Sometimes I get so tired that I faint.*"

(*5-year-old girl*)

"*Working at the crossroads is so hard in winter because it gets very cold and you can't light a fire to warm yourself.*"

(*7-year-old boy*)

"*When I go home at night, I don't know how to sleep; my whole body hurts; I feel like I am dead on my feet.*"

(*15-year-old boy*)

"*Sometimes I do not take off my shoes for 12 hours. When I do, I see my feet are blistered and sore.*"

(*14-year-old boy*)

**Inappropriate accommodation.**   Considering the economic and social conditions of Afghan families, they mostly live in poor and marginal neighbourhoods, considered high-risk neighbourhoods. Such improper living conditions encourage children to engage in high-risk behaviours such as buying and selling drugs. They often live in small houses, where they are forced to live with several families in the same room to lower the rent. This could significantly lower their quality of life. Many Afghan children grew up in large families with a large population and had to live in small rooms. This issue affected their health in all aspects, so even taking a bath was difficult. Some children said they sometimes had no chance to bathe for a whole month.

"*Most of our neighbours are drug dealers.*"

(8-year-old girl)

"*Our neighbourhood is very poor. There are many fights during the day. There are a lot of thefts. Overall, when we reach the street where we live, we feel terrified and insecure, and no one can stop us.*"

(*15-year-old boy*)

"*We live with the family of three of my uncles in an old 60-metre house.*"

(*11-year-old girl*)

"*Too often I can't sleep properly at night. There are ten of us in a small room.*"

(*12-year-old boy*)

"*It's our turn to take a bath. Sometimes we can't take a bath for several weeks.*"

(*9-year-old girl*)

**Medical and therapeutic problems.**   Another problem with child labourers was that they all did not have health insurance, and even those whose fathers were manual workers did not have insurance. Since the medical costs in Iran are very high for those with no insurance, most of these children, when sick, fail to visit a hospital. In some cases, when visiting hospitals and health care centres, some staff do not treat Afghan refugees very well. That is why they visit health centres significantly less. Another problem for Afghan child labourers was that they were mostly not vaccinated against diseases. Therefore, many were suffering from different diseases and experiencing many health issues.

"*When we get sick, if we feel terrible, we visit a doctor because our expenses are very high and we don't have any insurance.*"

(*8-year-old boy*)

"*Some doctors and nurses do not behave properly when they find out that I am an Afghan. Sometimes they insult us.*"

(*11-year-old boy*)

"*I was born in Iran because we came there illegally. Fearing that they would pick on us, I didn't go for any vaccinations, even those that were free.*"

(*12-year-old girl*)

**Health threat.**   Afghan child labourers were prone to many risks due to their conditions. Since they spent most of their time outdoors during the day, these children were more exposed to COVID-19 than others. Also, some who were scavenging were exposed to hepatitis and other infectious diseases due to the cuts that occurred to their skin. Since some of these children were sexually assaulted and experienced unsafe sex, they could be exposed to HIV and other infectious diseases. Considering that many children spent most of their time outside, they did not have the opportunity to return home for lunch or dinner. Most of the time, they did not have adequate nutrition and suffered from malnutrition. Child labourers spend most of their time outside exposed to air pollution, which sometimes causes them to be poisoned and can endanger their health. Moreover, working in the street all the time exposes them to accidents.

"*Since the beginning of the COVID-19 pandemic, because I am always out there in the streets, I have been infected several times. It bothered me a lot because I could not even visit a doctor.*"

(*9-year-old boy*)

"*One or two of my friends got hepatitis; I think maybe because they cut their hands while scavenging several times with broken glass.*"

(*15-year-old boy*)

"*I eat out most days; I mostly eat cold food.*"

(*6-year-old girl*)

"*The days when the air is polluted, I find it really hard. There have been several times that I fainted.*"

(*9-year-old girl*)

"*So far, I have had several accidents and been hit by a car; once I broke a leg and had to come to work with a broken leg.*"

(*6-year-old boy*)

## 3. Social challenges

Due to being refugees and suffering a very low socioeconomic status, Afghan child labourers faced many social problems such as neglected childhood, dual identity, educational limitations, a lack of sufficient social support, social isolation, and social humiliation.

**Neglected childhood.**   Most children spent all their time working outdoors. Therefore, they did not have a chance to live as children and play, and this issue put a lot of psychological pressure on them. Also, Afghan girls were forced to marry at a young age. Even some were forced to marry men who were very different in age in exchange for money. In fact, it can be said that most of these children had forgotten the world of their childhood and entered the world of adulthood; therefore, they suffered a lot of pressure and may continue to face many injuries that threaten their health.

"*When I'm at home and I'm unemployed, I watch more TV because I don't have any toys to entertain myself with; sometimes I feel like I'm not a kid at all.*"

(*7-year-old boy*)

"*All my time is spent in the streets; that's why I don't get any chance to play or go to the park. I don't remember the last time I went to the park and played. I want to be a child and play and be like other children.*"

(*11-year-old boy*)

"*I was 12 years old when I married a man who was much older than me. I lived with him for a year, and then he returned me to my family.*"

(*15-year-old girl*)

**Dual identity.**   Some child labourers were born in Iran and had never been to Afghanistan even once. Therefore, they had no knowledge of Afghan culture, and there was this duality for children to consider themselves Iranian or Afghan, especially since they were not recognised as Iranians in Iran and were still viewed as Afghans. In fact, Afghan children did not know whether to consider themselves Iranian or Afghan. They were born in Iran and spent their whole lives there, but they were born to Afghan parents. Therefore, they had a duality between Iranian and Afghan identities. Also, most of these children did not have any identity documents, and many did not even know their age, which could affect their identity and self-knowledge.

"*I was born in Iran. I have never been to Afghanistan. I really don't know if I should call myself Iranian or Afghan.*"

(*11-year-old girl*)

" *don't even know how old I am; I don't have any birth certificate or document to show how old I am.*"

(*9-year-old girl*)

"*I was born in Iran and grew up here. Even my father was born in Iran, but no one recognises us as Iranians.*"

(*15-year-old boy*)

"*Not knowing how old you are and when your birthday is really bothers me.*"

(*8-year-old boy*)

**Educational limitations.**   Most Afghan child labourers were deprived of education. This deprivation was partly due to their illegal presence in Iran and partly due to their lack of any identification certificates, and their age was not known at all. Some other limitations were due to the small number of schools that were not allowed to enrol Afghan children. Some of these children lived in neighbourhoods where there was no school. Therefore, most of them were deprived of education. Some schools refused to enrol Afghan children because they thought that by enrolling Afghan children, the peace of the school would be disturbed and the other children's parents would be dissatisfied. Also, some children and even school principals did not have a proper view of Afghan children, so Afghan children faced a lot of harassment in schools.

"*My family and I are here illegally; that's why I can't go to school; I don't even know how to read and write.*"

(*13-year-old girl*)

"*We live in an area where there is no school that enrols us Afghans; that's why my father didn't let me go to school because if I did, I would have to move away from home.*"

(*8-year-old girl*)

"*My father and I went to enrol at a school. The school principal insulted us and said, This is not the place for you Afghans.*"

(*10-year-old boy*)

"*I went to school for a couple of years, but the teachers and administrators treated me so badly that I stopped going there.*"

(*11-year-old boy*)

"*Many children insulted me in class and even beat me. I didn't care about school anymore.*"

(*13-year-old boy*)

**Inadequate social support.**   Despite the many health and social problems that Afghan child labourers had, there were few institutions or social organisations that would support them. When many of the institutions working to support children realised these children were Afghans, they refused to support them and said they should go where they belonged. Also,

despite the many dangers they face in their daily lives, child labourers need education so that they can better cope with the existing conditions, but they have not received such training.

> "*If something happens to me, I have to deal with it myself. There is nowhere to go to defend my rights.*"
>
> (*14-year-old boy*)
>
> "*Sometimes the municipality interferes and organises a class for us that tells us to behave in a way that we will be less injured, but these classes are very few in number, and most of the time no one attends them.*"
>
> (*11-year-old girl*)
>
> "*Until now, no workshop or class has been organised for me to tell me how to protect myself in the streets.*"
>
> (*12-year-old girl*)
>
> "*I have been beaten many times, but there is no place to go to complain. If I go to the police, they will arrest me because I am an illegal resident in Iran.*"
>
> (*13-year-old boy*)

**Social isolation.** Most Afghan children admitted that they did not have much contact with others or even their families. Even though they had been in Iran for several decades, they were not yet able to socialise with Iranian families. Their social affairs were only with other Afghan families living in Iran. Therefore, most of these children were socially isolated. Also, Afghan children working basically at crossroads constantly change their place of work because of the fear of municipal officials. So, they do not have a fixed place to stay. They fail to make good friends, which makes them further isolated at work. Although Afghan child labourers speak Persian and are conversant with Iranian child labourers, they can be distinguished by their distinct accent and somehow different appearance. Even their families do not tend to make friends with Afghan children. This problem aggravates Afghan children's social isolation.

> "*I have almost no friends; I do not get close to anyone. I often like to have a relationship with others, but as soon as they see I am Afghan, hardly anyone is willing to make friends with me.*"
>
> (*8-year-old girl*)
>
> "*I made friends with an Iranian boy at school. One day he invited me to his house, but his parents treated me very badly as soon as they saw that I was Afghan, and I heard them telling their son that he could not invite me home.*"
>
> (*12-year-old boy*).
>
> "*Our family only socialises with our relatives in Iran, who are also in other cities and rarely visit us.*"
>
> (*6-year-old girl*)
>
> "*When I'm not working, I'm always at home; I don't have anywhere to go. We don't go to parties at anyone's house, and no one comes to visit us either.*"

(*10-year-old boy*)

"*If I want to make friends with someone, I have to change my clothes and go to another place to sell things because of the municipal officials; that's why I don't have any close friends.*"

(*14-year-old boy*)

**Social humiliation.**   Afghan children said that they were constantly insulted in the streets and named impolite things in a humiliating manner. Considering that Afghans are usually willing to work in Iran for less money, some Iranians do not have a positive attitude towards them and say their presence ruins the labour market and causes the employer to pay the employees less. So they don't treat Afghan workers well.

"*Because of my appearance, it is obvious that I am Afghan; that's why they keep insulting me very often.*"

(*14-year-old boy*)

"*I was working somewhere, and as soon as they found out I am Afghan, they began to insult me and call me by different insulting names before I stopped working there anymore.*"

(*13-year-old boy*).

"*Sometimes when I'm in the streets, I get insulted a lot. They use very abusive words to call me or my family. I can't do anything about it.*"

(*10-year-old boy*)

## Discussion

This research was conducted with the aim of analysing the challenges facing Afghan child labourers in Tehran using a qualitative approach. The findings showed that Afghan child labourers face many psychological, health, and social challenges that threaten their health. An important psychological challenge facing Afghan child labourers was the experience of harassment and abuse, which was consistent with the related body of literature [38, 39]. In a study by Moayad et al., which was conducted in Iran, it was reported that more than 77% of Iranian child labourers had experienced one type of harassment [40], and the rate of harassment could be higher in Afghan children as they were refugees. The experience of sexual abuse among child labourers has been reported in several studies [41, 42]. Since Afghan children are refugees and some Iranians do not hold a positive attitude towards them, they are constantly prone to harassment and abuse, and since most of them are illegally present in Iran, they cannot follow up on this. The victims of harassment or abuse were forced to keep silent and cover up what happened to them, which can cause more harassment and abuse in the future. The presence of violence and abuse can happen to both sexes, but considering the cultural contexts that exist among Iranians and Afghans, it can have more destructive consequences for girls. Therefore, female child labourers usually spent fewer hours in the streets. It is suggested that free counselling courses be held to prevent violence against Afghan child labourers in order to reduce these instances of violence as much as possible. Additionally, these courses should provide training on what actions to take after experiencing violence to minimise the consequences they may face.

Another psychological challenge for Afghan child labourers was having negative feelings, which was consistent with previous studies [43]. Aransiola and Justus (2018), in their study conducted in Brazil, reported that working as a child can lead to depression in adulthood [44]. Afghan child labourers are in very fragile conditions, and they always live with fear and anxiety due to their illegal presence in Iran. The experience of many types of harassment has produced distrust in them, so they can no longer trust anyone. Also, Afghan child labourers compare their lives with other children who enjoy better conditions, and this comparison can eventually cause jealousy or even aggression towards the peers.

High-risk behaviours were another challenge facing Afghan child labourers, which is consistent with the findings of many of the previous studies [28, 45, 46]. Abate et al. reported in some research in Ethiopia that alcohol consumption is a high-risk behaviour among child labourers [19]. Living in poor and dangerous areas causes children to get to know and learn about all kinds of crimes at an early age and sometimes rely on those crimes as a source of income. So, some of them were abused by criminals to buy and sell drugs. Some other child labourers lived in groups under the supervision and even ownership of someone special. They lived as a group in dangerous and crowded areas, sometimes prone to risky sexual behaviours, which can, in the future, make them more susceptible to sexually transmitted diseases such as AIDS and hepatitis. Also, some children experienced stealing things from people, which can be due to a sense of jealousy and hatred towards people of a higher socioeconomic status.

Family detachment was another mental challenge that was new in the present study. Working on the street far from home can damage the relationship between the working child and the parents. It can also reduce parental supervision over them, which can threaten their health and prospects. Also, the decision about how to spend the income they have earned can cause tension in the family, which, if not managed properly, can have bad consequences for the child.

Physical problems comprised another health challenge facing child labourers, which is consistent with Habib et al.'s study conducted among Syrian refugee child labourers in Lebanon [47]. The results of a study conducted in India show that child labourers have many physical problems such as sore eyes, headaches, nausea, diarrhoea, backaches, and skin diseases [48]. Many hours of being in the street in hot and cold seasons without adequate protective clothes, rest, and improper nutrition can cause many physical problems, such as physical weakness, sunburn, skin problems, digestive problems, colds, backaches, sore toes, and so on, which should be treated in children. It is suggested that child labourers undergo periodic medical examinations so that, in the event of any physical problems, they can be referred to a specialist and preventive measures can be taken to mitigate the severity of illnesses.

Another finding of the present research was the children's inappropriate accommodation, which was a new finding in this study. Refugees usually live on the outskirts of cities and in slums, which can threaten their health. Afghan child labourers' accommodation was improper and unsafe in two ways. On the one hand, they lived in slums replete with crime, and on the other hand, they lived in tiny houses where many people were forced to live together in small rooms and use a common bathroom, which adversely affected their health. That was why some participants had not taken a bath for a month.

Medical and therapeutic problems were also extracted from the present study, which is consistent with the findings reported by Yoosefi Lebni et al. [49]. Afghan children are deprived of insurance services due to their illegal presence in Iran, and since the medical costs are very high in Iran, they mostly refrain from going to the hospital when they are sick and retreat to self-medication. Also, for fear of being sent back to Afghanistan, they refrain from going to health centres for vaccination, which can cause many problems for their health in the future. Among the other problems facing Afghan children, it was reported that the occasional

inappropriate behaviour of the medical staff towards Afghan refugees can affect Afghans willingness to see a doctor. Considering the points mentioned, it is recommended that health insurance be provided for Afghan child labourers so that they can easily access medical facilities.

The health threat was another challenge facing Afghan child labourers. Their health is constantly threatened due to their working conditions. They spend most of the day outdoors, which has consequences. They are less aware of the health protocols. They can hardly receive any health services due to the high costs. They do not follow the health protocols, and this can expose them to COVID-19 and other infections. As mentioned in previous studies, refugees are among the populations at high risk of COVID-19 infection [50–52]. Children who engage in scavenging may get diseases such as hepatitis due to the frequent cuts on their hands. As reported in a work of research by Hosseinpour et al., cuts are one of the most common problems with child labourers [53]. Forceful and high-risk sex can also increase the rate of infection with sexually transmitted diseases. Another study by Foroughi et al. showed that children working in the streets are more susceptible to HIV and hepatitis B and C than the general population [15]. Also, their constant presence in the street exposes them to many accidents, especially car accidents, which can lead to nutritional problems in the long run and even malnutrition. The results of some other studies confirm this issue too. Malnutrition and problems such as vitamin deficiency and anaemia are other problems the child labourers suffer from [45].

Neglected childhood was another problem facing Afghan child labourers. These children spend all their time in the streets and do not have any time to play. Even if they had, due to poverty, they did not have any toys to play with and entertain themselves with. Therefore, it can be concluded that their childhood has ended very soon and they have entered the adult world sooner than they should. This issue is more critical for female children, as some are forced to get married at a very young age. Most of these marriages happen by force and for financial reasons. Having a wide age gap with the husband exposes them to sexual harassment and other forms of abuse.

In addition to health-related and psychological problems, Afghan children also faced many social challenges, one of which was dual identity. In fact, most child labourers were born and grew up in Iran and knew nothing about Afghanistan; however, they were still considered refugees and had not been able to fully integrate into Iranian society. This issue created a kind of dual identity. They did not know whether they really considered themselves Afghan or Iranian. They mostly liked to consider themselves Iranian and tried to speak exactly like Iranians as much as they could so that they were not recognised as foreigners. In Iranian society, they were still considered Afghans.

Another challenge facing Afghan child labourers was their educational limitations. In most previous studies, dropping out of school has been reported as a common experience for child labourers [54, 55]. In a study in Indonesia, improving the quality of school service provision can be a solution to the elimination of child labour [56]. Many children were denied education due to the lack of ID certificates and their illegal presence in Iran. Also, the schools for Afghan children were limited in number, and many children refused to go to school due to the distance. Also, the negative attitude of some school principals towards Afghan children was the reason for avoiding the admission of Afghan children to school. Increasing the number of schools dedicated to Afghan children and gradually integrating them with Iranian schools can facilitate the integration of Afghan and Iranian children.

Another social challenge was the lack of sufficient social support. This finding was new to the current study. In Iran, social organisations accountable for child labourers usually do not pay attention to the child labourers of Afghanistan. A limited number of these organisations were supporting Afghan children, and they did not have a coherent plan to promote these

children's health. Previous studies showed that social support for child labourers can increase their self-confidence and cohesion and can prevent violence [57]. Another social problem for Afghan labour children was social isolation. The low-level social skills of child labourers in communication have been reported in previous studies [58]. Most child labourers in Afghanistan had limited social relations, largely due to their inability to communicate due to non-acceptance by Iranians and partly due to their working conditions. They are constantly changing their workplace for fear of arrest by the officers, and because they are not in a fixed place, they cannot make many friends. Another social challenge that Afghan children faced on a daily basis was social humiliation. Although it has been several decades since Afghans immigrated to Iran, they still have not been able to integrate into Iranian society. So many times, Afghan child labourers were insulted in the streets and called abusive names, and they were humiliated very often.

## Strengths and limitations

The results of the present study are important to benefit from the human capital of refugees for the national interests of the country based on humanitarian standards to prevent serious social and security harms. This paper is part of a larger research project that explores the challenges of Afghan child labourers broadly and based on real experience. Therefore, the results of the present study can provide valuable information to policymakers and planners so that they can take effective steps to improve children's health and make appropriate and evidence-based decisions. Another strength of this study was the inclusion of experts from different disciplines in conducting the research, which addressed child labourers' challenges from a broader perspective. Another strength of this research was that, unlike previous studies that emphasised child labourers' physical and psychological problems, this study also paid attention to their social challenges. However, this research, like other studies, faced certain limitations. One of the most important limitations was that some Afghan children were not really willing to be interviewed and were worried that recording the interview would become problematic. Yet, attempts were made to fully explain the interview process and how to publish the results, as well as the commitment to use the findings only for research purposes. Fully informed consent was given to participate in the interview and do the audio recording. Furthermore, some parents of the children involved in the study were unwilling to have their children participate in the interviews due to concerns about their names and addresses being revealed, which could lead to their identification. However, the authors managed to gain their consent by explaining the research process and assuring them that no names or addresses of the participants would be disclosed.

## Conclusion

The present findings showed that the child labourers of Afghanistan are facing many mental, health-related, and social challenges. Any attempt to improve their health requires intervention at different levels. At the individual level, it is possible to meet their health needs and make them aware of the dangers of working in the street through regular physical examinations by holding different training workshops such as violence prevention, anger control, prevention of risky behaviours, prevention of infectious diseases such as hepatitis, and strengthening self-confidence. Also, at the social and familial level, interventions can be made by adopting effective policies at the macro level, such as providing accommodation in safer neighbourhoods, health insurance for child labourers, allowing them to study at school, preventing dropouts, and strengthening their social and interpersonal relations to socialise and improve the health of Afghan child labourers. Training programmes and workshops can help with this.

## Supporting information

**S1 File. This is the full semi-structured interview script.**
(DOCX)

**S1 Table. This is the complete list of study themes and representative quotes.** Each quote within each theme is from a different study participant.
(DOCX)

## Author Contributions

**Conceptualization:** Arash Ziapour, Javad Yoosefi Lebni.

**Data curation:** Ahmad Ahmadi, Javad Yoosefi Lebni.

**Formal analysis:** Arash Ziapour, Javad Yoosefi Lebni.

**Funding acquisition:** Arash Ziapour.

**Investigation:** Arash Ziapour, Fakhreddin Chaboksavar, Parisa Janjani.

**Methodology:** Arash Ziapour, Javad Yoosefi Lebni.

**Software:** Arash Ziapour, Neda Kianipour.

**Supervision:** Arash Ziapour, Javad Yoosefi Lebni.

**Writing – original draft:** Arash Ziapour, Javad Yoosefi Lebni.

**Writing – review & editing:** Arash Ziapour, Javad Yoosefi Lebni, Nafiul Mehedi.

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
