## [Decision Letter · Decision Letter 0]

24 Apr 2023

PONE-D-23-04890A Qualitative Study of the Health-Related Challenges of Afghan Child Laborers in TehranPLOS ONE

Dear Dr. ziapour,

Thank you for submitting your manuscript to PLOS ONE. After careful consideration, we feel that it has merit but does not fully meet PLOS ONE’s publication criteria as it currently stands. Therefore, we invite you to submit a revised version of the manuscript that addresses the points raised during the review process.

As you will see, the reviewers have highlighted a number of important issues that need to be addressed. In particular, a number of ethics issues are highlighted, which would be important to clarify. In general, I would advise to use the COREQ guidelines to ensure that you are reporting on all essential aspects of a qualitative study. Please submit your revised manuscript by Jun 08 2023 11:59PM. If you will need more time than this to complete your revisions, please reply to this message or contact the journal office at plosone@plos.org. Please include the following items when submitting your revised manuscript:A rebuttal letter that responds to each point raised by the academic editor and reviewer(s). You should upload this letter as a separate file labeled 'Response to Reviewers'.A marked-up copy of your manuscript that highlights changes made to the original version. You should upload this as a separate file labeled 'Revised Manuscript with Track Changes'.An unmarked version of your revised paper without tracked changes. You should upload this as a separate file labeled 'Manuscript'.

We look forward to receiving your revised manuscript.

Kind regards,

Rafael Van den Bergh

Academic Editor

PLOS ONE

Journal Requirements:

Additional Editor Comments:

Thank you for submitting your interesting manuscript to PLOS ONE for review. Two expert reviewers have provided their in-depth comments on your work; in addition, I would like to make the following suggestions from my side for your consideration:

* Overall, I would recommend having the manuscript reviewed by a native English speaker, as some confusion may be created by the language issues.

* I would suggest breaking the methods section into formal paragraphs, to enhance general readability

* The consent/assent process of the research needs to be described in more detail - as this study concerns a particularly vulnerable population of children, who cannot necessarily give informed consent by themselves, a detailed clarification is required. Were steps taken to protect participants in case particular vulnerabilities were observed?

* The study seems to extend well beyond only health aspects, and I would suggest rephrasing the title to reflect this.

* I would suggest that the authors take the opportunity of this study to speculate on particular solutions to a number of the challenges they describe. Which kind of protection mechanisms should be considered for this population, which stakeholders would need to be engaged? By providing some suggestions on this, the impact of the work may be enhanced.

Reviewers' comments:

Reviewer's Responses to Questions

**Comments to the Author**

1. Is the manuscript technically sound, and do the data support the conclusions?

Reviewer #1: Partly

Reviewer #2: Partly

2. Has the statistical analysis been performed appropriately and rigorously? 

Reviewer #1: N/A

Reviewer #2: N/A

3. Have the authors made all data underlying the findings in their manuscript fully available?

Reviewer #1: Yes

Reviewer #2: No

4. Is the manuscript presented in an intelligible fashion and written in standard English?

Reviewer #1: No

Reviewer #2: Yes

5. Review Comments to the Author

Reviewer #1: This article addresses an extremely important topic and deals with some of the most vulnerable people in modern society - child labourers. I have some significant questions associated with the research ethics that guided this project and the way these issues are addressed in the article. I also have other comments related to the content of the article but I am raising these ethical concerns first. The authors may be able to address these questions with ease and provide more detail in a revised version of the article.

Research ethics is addressed under the subheading of 'study design'. I know the details of institutional ethics were provided. However, I do have some concerns about how the data was collected and thus would like more information about ethics approval and informed consent. Participants were aged 15 years or below (including 8 children aged under 10) – how did they give informed consent to participate? Can such young children really given ‘consent’ to participate in this research? The section on ‘Ethical considerations’ gives some of details but the involvement of the guardian needs to be highlighted earlier on to reassure the reader. Was a guardian asked to consent on the child's behalf after both parties were given all the details of the research? What protections were put in place to avoid potential harm? Based on the information provided in the article, it sounds as though the researchers just walked up to children on the street and asked them to participate in interviews (alone with the interviewer) and then contacted guardians after recruitment. Is this correct? What did the researchers do after hearing the stories of abuse etc suffered by the children? Were any incentives (vouchers etc) given to encourage participation? Did any guardians refuse permission to participate? Given the vulnerability of the research participants, these questions need to be clearly addressed early on in the article so the reader feels comfortable with the article itself.

In addition to these queries, I also some more general comments about the article. Some English editing is required to remove ambiguities and inaccuracies in wording. Although the grammar is often correct, some errors in word choice make the meaning unclear at times. This editing needs to be done throughout the article. I have given some specific examples below.

Introduction

Some English editing is required throughout the article to clarity meaning and guide the reader more. There are occasional ambiguities that could be resolved through careful editing. For example, on page 8 of the PDF:

The sentence: Iran is a country with a vast majority of Afghan immigrants

This sentence is somewhat unclear – does it mean that most immigrants in Iran are from Afghanistan OR does it mean that a majority of the immigrants who leave Afghanistan go to Iran (rather than other countries)?

Is the term ‘immigrant’ being used to encompass refugees as well as people who may choose to leave Afghanistan? This term could be defined more clearly.

Another example from page 8 of the PDF is shown below:

The increasing rate of child labor is so high that it has affected both developed and developing societies.

The sentence is grammatically correct but the meaning is too general for the paragraph in which it appears. The next paragraph then goes on to define child laborers – this could be usefully done earlier in the article.

Another example from page 8 of the PDF is shown below:

There is ample evidence that the most common reasons for child laborers to live in streets are to escape the insulting punishment of parents, poverty and the abusive behavior of alcoholic parents.

The meaning of ‘insulting punishment’ is not clear in English. A term like ‘abusive behaviour’ would be clearer.

On page 8 of the PDF, the paragraph beginning with “The results of a study by Banstola et al., …” presents interesting literature review information about the health impacts of child labour in a variety of countries. This information needs to be embedded more within this section of the article. A general sentence introducing the paragraph could help – e.g. “Child labour has been linked with significant and long term health impacts….”

Results

This section is largely descriptive rather than analytical. Data could be more thoroughly analysed and grouped into themes. Some of the terminology also needs to be updated to reflect more empathetic ways of discussing the challenges these children may face. For example, ‘mental challenges’ is not very meaningful as a term. The discussion is more about the psychological impact of the range of abusive and traumatic experiences described. A lot of quotes from interviewees are included which lengthens the analysis but these are not really closely connected to the surrounding discussion (e.g. section under ‘neglected childhood’).

The other sections of the article could also be strengthened through care English editing and rewriting. Please note that the examples I have given are only a selection. The whole article needs careful rereading and editing for clarity.

The content of the Discussion and Conclusion sections will also be influenced by how the authors respond to my questions around research ethics. What protections were put in place for such incredibly vulnerable children? The role of the research team needs to be examined as well - what motivated the team to pursue this topic? How are the findings being fedback into the systems/organisations that are meant to protect these children (if there are no such protections then this ought to be highlighted as well.

Reviewer #2: Many thanks for giving me the opportunity to review the manuscript. Authors have conducted an interesting and valuable qualitative study to explore various challenges faced by Afghan child labourers in Tehran. The following are few points that need to be addressed before the acceptance of the manuscript.

- Title of the paper is very much focused on health-related challenges, whereas findings revealed that there are other types of challenges as well such as mental and social. It will be useful if authors consider revising the title.

-Authors have mentioned that data are available without restriction but haven’t shared details related to data accessibility. Please mention how data can be accessed either as a supplementary file or via any repository etc.

Methods:

Study design:

- Please add background/specialty of the research team who developed research questions.

- Please also briefly add some details about the process of the pilot phase.

- Study procedure is missing. Please follow the PLOS ONE structure for the development of the manuscript.

Results section:

- Too many quotes have been included to support the findings. It would be helpful if authors could identify few key relevant quotes and include them under each theme

- To ensure consistency and avoid confusion, please add numbers to the main themes.

- Also, the sub-theme 'neglected childhood' can be placed under social or mental challenges instead of health challenges.

Discussion:

-It would be interesting to add in discussion the challenges authors face during data collection with child laborers

6. PLOS authors have the option to publish the peer review history of their article (what does this mean?). If published, this will include your full peer review and any attached files.

Reviewer #1: No

Reviewer #2: **Yes: **Bushra Khan

---

## [Author Response · Author response to Decision Letter 0]

4 Feb 2024

We, the authors of the article, would like to thank the Editor and the honorable reviewers for their valuable comments that have helped us to improve the quality of this article. We did our best to consider the comments of the reviewers in the manuscript. The authors are very eager to publish the current article in PLOS ONE

Thank you for submitting your interesting manuscript to PLOS ONE for review. Two expert reviewers have provided their in-depth comments on your work; in addition, I would like to make the following suggestions from my side for your consideration:

* Overall, I would recommend having the manuscript reviewed by a native English speaker, as some confusion may be created by the language issues.

Response: It was changed. 

* I would suggest breaking the methods section into formal paragraphs, to enhance general readability

Response: It was changed

* The consent/assent process of the research needs to be described in more detail - as this study concerns a particularly vulnerable population of children, who cannot necessarily give informed consent by themselves, a detailed clarification is required. Were steps taken to protect participants in case particular vulnerabilities were observed?

Response: No official measures were taken to protect the participants, but in some cases, participants who had been abused or had substance use were referred to a psychologist for free treatment and counseling.

* The study seems to extend well beyond only health aspects, and I would suggest rephrasing the title to reflect this.

Response: It was changed

* I would suggest that the authors take the opportunity of this study to speculate on particular solutions to a number of the challenges they describe. Which kind of protection mechanisms should be considered for this population, which stakeholders would need to be engaged? By providing some suggestions on this, the impact of the work may be enhanced.

Response: Thank you for your valuable comment. Solutions for some of the challenges have been added to the discussion section.

Reviewer #1: This article addresses an extremely important topic and deals with some of the most vulnerable people in modern society - child labourers. I have some significant questions associated with the research ethics that guided this project and the way these issues are addressed in the article. I also have other comments related to the content of the article but I am raising these ethical concerns first. The authors may be able to address these questions with ease and provide more detail in a revised version of the article.

Response: Thank you for your valuable comment.

Research ethics is addressed under the subheading of 'study design'. I know the details of institutional ethics were provided. However, I do have some concerns about how the data was collected and thus would like more information about ethics approval and informed consent. Participants were aged 15 years or below (including 8 children aged under 10) – how did they give informed consent to participate? Can such young children really given ‘consent’ to participate in this research? The section on ‘Ethical considerations’ gives some of details but the involvement of the guardian needs to be highlighted earlier on to reassure the reader. Was a guardian asked to consent on the child's behalf after both parties were given all the details of the research? What protections were put in place to avoid potential harm? Based on the information provided in the article, it sounds as though the researchers just walked up to children on the street and asked them to participate in interviews (alone with the interviewer) and then contacted guardians after recruitment. Is this correct? 

Response: . The process of obtaining informed consent was as follows: Upon approaching Afghan child laborers, the research process was thoroughly explained to the participants, and the consent form was read to them and fully explained. It was emphasized that participation in the study was voluntary, and they had the option to discontinue the interview at any time or choose a different time and place. Additionally, they were asked to contact one of their parents so that the researchers could meet with them and gain their consent for their children's participation in the study, providing a detailed explanation of the research process and how the results would be disseminated. During the interview, only the researcher and the children were present, and the parents were not present at the interview location. For some participants, a meal was prepared and provided to them. Additionally, for some others, the researcher purchased some of the items they were selling

What did the researchers do after hearing the stories of abuse etc suffered by the children? 

Response:No official measures were taken to protect the participants, but in some cases, participants who had been abused or had substance use were referred to a psychologist for free treatment and counseling..

Were any incentives (vouchers etc) given to encourage participation?

Response: For some participants, a meal was prepared and provided to them. Additionally, for some others, the researcher purchased some of the items they were selling

 Did any guardians refuse permission to participate? Given the vulnerability of the research participants, these questions need to be clearly addressed early on in the article so the reader feels comfortable with the article itself.

Response:Yas. Three of the parents did not allow their children to participate in the research at first, but by referring to them again and explaining the research process, their consent was obtained.

In addition to these queries, I also some more general comments about the article. Some English editing is required to remove ambiguities and inaccuracies in wording. Although the grammar is often correct, some errors in word choice make the meaning unclear at times. This editing needs to be done throughout the article. I have given some specific examples below.

Response: It was changed

Introduction

Some English editing is required throughout the article to clarity meaning and guide the reader more. There are occasional ambiguities that could be resolved through careful editing. For example, on page 8 of the PDF:

The sentence: Iran is a country with a vast majority of Afghan immigrants

This sentence is somewhat unclear – does it mean that most immigrants in Iran are from Afghanistan OR does it mean that a majority of the immigrants who leave Afghanistan go to Iran (rather than other countries)?

Response: It was changed

Is the term ‘immigrant’ being used to encompass refugees as well as people who may choose to leave Afghanistan? This term could be defined more clearly.

Response: Thank you for your valuable comment. In the entire text of the article, the word refugee was used instead of the word immigrant, and the correction was made.

Another example from page 8 of the PDF is shown below:

The increasing rate of child labor is so high that it has affected both developed and developing societies.

The sentence is grammatically correct but the meaning is too general for the paragraph in which it appears. The next paragraph then goes on to define child laborers – this could be usefully done earlier in the article.

Response: It was changed

Another example from page 8 of the PDF is shown below:

There is ample evidence that the most common reasons for child laborers to live in streets are to escape the insulting punishment of parents, poverty and the abusive behavior of alcoholic parents.

The meaning of ‘insulting punishment’ is not clear in English. A term like ‘abusive behaviour’ would be clearer.

Response: It was changed

On page 8 of the PDF, the paragraph beginning with “The results of a study by Banstola et al., …” presents interesting literature review information about the health impacts of child labour in a variety of countries. This information needs to be embedded more within this section of the article. A general sentence introducing the paragraph could help – e.g. “Child labour has been linked with significant and long term health impacts….”

Response: It was changed

Results

This section is largely descriptive rather than analytical. Data could be more thoroughly analysed and grouped into themes. Some of the terminology also needs to be updated to reflect more empathetic ways of discussing the challenges these children may face. For example, ‘mental challenges’ is not very meaningful as a term. The discussion is more about the psychological impact of the range of abusive and traumatic experiences described. A lot of quotes from interviewees are included which lengthens the analysis but these are not really closely connected to the surrounding discussion (e.g. section under ‘neglected childhood’).

Response: It was changed

The other sections of the article could also be strengthened through care English editing and rewriting. Please note that the examples I have given are only a selection. The whole article needs careful rereading and editing for clarity.

Response: It was changed

The content of the Discussion and Conclusion sections will also be influenced by how the authors respond to my questions around research ethics. What protections were put in place for such incredibly vulnerable children? The role of the research team needs to be examined as well - what motivated the team to pursue this topic? How are the findings being fedback into the systems/organisations that are meant to protect these children (if there are no such protections then this ought to be highlighted as well.

Response:No official measures were taken to protect the participants, but in some cases, participants who had been abused or had substance use were referred to a psychologist for free treatment and counseling..

The authors of the article, particularly the first author and the corresponding author, have been conducting research on issues related to vulnerable groups such as women and children for years and have published in reputable journals. One of the reasons for investigating this phenomenon is their life experience in Tehran and witnessing the repeated violence against Afghan child laborers on the streets.

The findings of the article will be presented at various conferences and seminars related to children, and the research results will also be shared with organizations working in the field of child labor and migration. This will help them become familiar with the challenges faced by Afghan child laborers and enable them to choose strategies to reduce this social harm.

Reviewer #2: Many thanks for giving me the opportunity to review the manuscript. Authors have conducted an interesting and valuable qualitative study to explore various challenges faced by Afghan child labourers in Tehran. The following are few points that need to be addressed before the acceptance of the manuscript.

Response: : Thank you for your valuable comment.

- Title of the paper is very much focused on health-related challenges, whereas findings revealed that there are other types of challenges as well such as mental and social. It will be useful if authors consider revising the title.

Response: It was changed

-Authors have mentioned that data are available without restriction but haven’t shared details related to data accessibility. Please mention how data can be accessed either as a supplementary file or via any repository etc.

Response: data are available by contacting the corresponding author.

Methods:

Study design:

- Please add background/specialty of the research team who developed research questions.

Response:. The research questions were designed by the authors of the article who have expertise in various fields including sociology, psychology, health education, and others.

- Please also briefly add some details about the process of the pilot phase.

Response: Initially, based on a review of previous research studies, the authors of the article collaboratively designed four interview questions for the participants. These questions were then experimentally implemented with four participants who met the inclusion criteria for the study. Subsequently, the interviews were analyzed, one question was removed, and three additional questions were added to the interviews. Finally, the interview guide was finalized with six final questions.

- Study procedure is missing. Please follow the PLOS ONE structure for the development of the manuscript.

Response: It was changed

Results section:

- Too many quotes have been included to support the findings. It would be helpful if authors could identify few key relevant quotes and include them under each theme

Response: It was changed

- To ensure consistency and avoid confusion, please add numbers to the main themes.

Response: It was changed

- Also, the sub-theme 'neglected childhood' can be placed under social or mental challenges instead of health challenges.

Response: 'neglected childhood' be placed under social challenges

Discussion:

-It would be interesting to add in discussion the challenges authors face during data collection with child laborers 

Response: Thank you for your valuable comment. Some concerns and limitations that the researchers encountered are mentioned in the strengths and limitations section of the study.

---

## [Decision Letter · Decision Letter 1]

7 May 2024

PONE-D-23-04890R1A Qualitative Study on the challenges of afghan child labourers in TehranPLOS ONE

Dear Dr. ziapour,

Thank you for submitting your manuscript to PLOS ONE. After careful consideration, we feel that it has merit but does not fully meet PLOS ONE’s publication criteria as it currently stands. Therefore, we invite you to submit a revised version of the manuscript that addresses the points raised during the review process.

We look forward to receiving your revised manuscript.

Kind regards,

Aloysius Odii, PhD

Academic Editor

PLOS ONE

Reviewers' comments:

Reviewer's Responses to Questions

**Comments to the Author**

1. If the authors have adequately addressed your comments raised in a previous round of review and you feel that this manuscript is now acceptable for publication, you may indicate that here to bypass the “Comments to the Author” section, enter your conflict of interest statement in the “Confidential to Editor” section, and submit your "Accept" recommendation.

Reviewer #2: All comments have been addressed

Reviewer #3: All comments have been addressed

2. Is the manuscript technically sound, and do the data support the conclusions?

Reviewer #2: Yes

Reviewer #3: Yes

3. Has the statistical analysis been performed appropriately and rigorously? 

Reviewer #2: N/A

Reviewer #3: Yes

4. Have the authors made all data underlying the findings in their manuscript fully available?

Reviewer #2: Yes

Reviewer #3: No

5. Is the manuscript presented in an intelligible fashion and written in standard English?

Reviewer #2: Yes

Reviewer #3: Yes

6. Review Comments to the Author

Reviewer #2: (No Response)

Reviewer #3: The study is an interesting one and the authors' made efforts at following standard academic research procedures to investigate the issue under discourse. I carefully read through the manuscript and I found it academically stimulating and contributory to the discourse and knowledge of child welfare policy and protection.

The authors followed the comments of the earlier reviewers, and implemented same in the manuscript which helped to improve the quality of the manuscript. Regarding making all data underlying the findings in the manuscript fully available, the authors explained that "The data are not publicly available due to their containing information that could compromise the privacy of research participants as the data contain potentially identifying child labourers information". However, the authors further stated that "Interview guide and Complete list of study themes and representative quotes is included as supplementary material. It is required that all researchers who wanted to work on this data submit their data request access". The position of the authors are ethical and should be respected.

Whereas I consider the manuscript standard enough for publication in its current form, there are two key observations I made. 1. Page 1 line 1 under abstract, the last in the sentence should be "refugees" in line with the subject matter (Afghan child labourers...).

2. Page 5 line 6 under ethical considerations, the sentence that started with "also", should be in initial Cap (Also,...).

The manuscript has therefore improved to meet the journal's standard for publication after the minor corrections.

7. PLOS authors have the option to publish the peer review history of their article (what does this mean?). If published, this will include your full peer review and any attached files.

Reviewer #2: **Yes: **Bushra Khan

Reviewer #3: No

---

## [Author Response · Author response to Decision Letter 1]

11 May 2024

Dear Editor and Reviewers,

We appreciate the interest that the editor and reviewers have taken in our manuscript and the constructive comments they have given. We have addressed the concerns of the reviewers and provided a point-by-point response to the comments. Additionally, we have highlighted the modifications in the paper.

Thank you again for consideration of our revised manuscript.

Sincerely,

Reviewers' comments:

Reviewer's Responses to Questions

Comments to the Author

1. If the authors have adequately addressed your comments raised in a previous round of review and you feel that this manuscript is now acceptable for publication, you may indicate that here to bypass the “Comments to the Author” section, enter your conflict of interest statement in the “Confidential to Editor” section, and submit your "Accept" recommendation.

Reviewer #2: All comments have been addressed

Reviewer #3: All comments have been addressed

Reviewer #2: (No Response)

Reviewer #3: The study is an interesting one and the authors' made efforts at following standard academic research procedures to investigate the issue under discourse. I carefully read through the manuscript and I found it academically stimulating and contributory to the discourse and knowledge of child welfare policy and protection.

The authors followed the comments of the earlier reviewers, and implemented same in the manuscript which helped to improve the quality of the manuscript. Regarding making all data underlying the findings in the manuscript fully available, the authors explained that "The data are not publicly available due to their containing information that could compromise the privacy of research participants as the data contain potentially identifying child labourers information". However, the authors further stated that "Interview guide and Complete list of study themes and representative quotes is included as supplementary material. It is required that all researchers who wanted to work on this data submit their data request access". The position of the authors are ethical and should be respected.

Whereas I consider the manuscript standard enough for publication in its current form, there are two key observations I made. 1. Page 1 line 1 under abstract, the last in the sentence should be "refugees" in line with the subject matter (Afghan child labourers...).

2. Page 5 line 6 under ethical considerations, the sentence that started with "also", should be in initial Cap (Also,...). The manuscript has therefore improved to meet the journal's standard for publication after the minor corrections.

Response:we have highlighted the modifications in the paper.

Thank you again for consideration of our revised manuscript.

Sincerely,

---

## [Editor Report · Decision Letter 2]

16 Jun 2024

A Qualitative Study on the challenges of afghan child labourers in Tehran

PONE-D-23-04890R2

Dear Dr. ziapour,

We’re pleased to inform you that your manuscript has been judged scientifically suitable for publication and will be formally accepted for publication once it meets all outstanding technical requirements.

Kind regards,

Aloysius Odii, PhD

Academic Editor

PLOS ONE

---

## [Editor Report · Acceptance letter]

4 Jul 2024

PONE-D-23-04890R2 

PLOS ONE

Dear Dr. Ziapour, 

I'm pleased to inform you that your manuscript has been deemed suitable for publication in PLOS ONE. Congratulations! Your manuscript is now being handed over to our production team.

Kind regards, 

on behalf of

Dr. Aloysius Odii 

Academic Editor

PLOS ONE